# Beyond the Umbrella: A Systematic Review of the Interventions for the Prevention of and Reduction in the Incidence and Severity of Ovarian Hyperstimulation Syndrome in Patients Who Undergo In Vitro Fertilization Treatments

**DOI:** 10.3390/ijms241814185

**Published:** 2023-09-16

**Authors:** Stefano Palomba, Flavia Costanzi, Scott M. Nelson, Aris Besharat, Donatella Caserta, Peter Humaidan

**Affiliations:** 1Unit of Gynecology, Sant’Andrea Hospital, Department of Surgical and Medical Sciences and Translational Medicine, Sapienza University of Rome, 00189 Rome, Italy; costanzi.flaviai@gmail.com (F.C.); besharataris@gmail.com (A.B.); donatella.caserta@uniroma1.it (D.C.); 2School of Medicine, University of Glasgow, Glasgow G12 8QQ, UK; scott.nelson@glasgow.ac.uk; 3Population Health Sciences, Bristol Medical School, University of Bristol, Bristol BS1 3NY, UK; 4The Fertility Partnership, Oxford OX4 2HW, UK; 5The Fertility Clinic, Skive Regional Hospital, Faculty of Health, Aarhus University, Aarhus C, 8000 Aarhus, Denmark; peter.humaidan@midt.rm.dk

**Keywords:** assisted reproductive technologies, ART, complications, in vitro fertilization, ovarian hyperstimulation syndrome, OHSS, systematic review

## Abstract

Ovarian hyperstimulation syndrome (OHSS) is the main severe complication of ovarian stimulation for in vitro fertilization (IVF) cycles. The aim of the current study was to identify the interventions for the prevention of and reduction in the incidence and severity of OHSS in patients who undergo IVF not included in systematic reviews with meta-analyses of randomized controlled trials (RCTs) and assess and grade their efficacy and evidence base. The best available evidence for each specific intervention was identified, analyzed in terms of safety/efficacy ratio and risk of bias, and graded using the Oxford Centre for Evidence-Based Medicine (CEBM) hierarchy of evidence. A total of 15 interventions to prevent OHSS were included in the final analysis. In the IVF population not at a high risk for OHSS, follitropin delta for ovarian stimulation may reduce the incidence of early OHSS and/or preventive interventions for early OHSS. In high-risk patients, inositol pretreatment, ovulation triggering with low doses of urinary hCG, and the luteal phase administration of a GnRH antagonist may reduce OHSS risk. In conclusion, even if not supported by systematic reviews with homogeneity of the RCTs, several treatments/strategies to reduce the incidence and severity of OHSS have been shown to be promising.

## 1. Introduction

Ovarian hyperstimulation syndrome (OHSS) is one of the major complications of ovarian stimulation with gonadotropins, particularly in in vitro fertilization (IVF) cycles. Its actual incidence in real-world populations has been difficult to define due to poor case ascertainment, but mild OHSS has been estimated to affect one-third of cycles [1]. In many cases, OHSS is caused by an excessive response to ovarian stimulation, even if in some cases it may be considered an idiosyncratic reaction to gonadotropins. Spontaneous OHSS cases not related to ovarian stimulation drugs have been also described [2,3].

To date, the molecular pathophysiology of OHSS has not been fully elucidated, but several distinct molecular pathways have been shown to be critical in the development of the range of clinical sequelae [4,5,6]. The first of these is human chorionic gonadotropin (hCG) due to its substantially greater biological activity than luteinizing hormone (LH) with respect to receptor affinity and half-life initiates many of the downstream events [7]. Other molecular pathways, independent or related to hCG receptor activation, include abnormal follicle expression of inflammatory cytokines [8], activation of the angiotensin–renin system [9], and an increased expression of vascular endothelial growth factor (VEGF) [10]. VEGF hyperactivation is associated with a massive increase in vascular permeability with the consequent transfer of liquids in the third space [10]. Other molecular pathways may independently induce hyperpermeability and/or potentiate the VEGF expression and activity, such as the activation of the kallikrein–kinin system [11] or the transforming growth factor (TGF)-β1-mediated regulation of SPARC in human granulosa cells [12] or the release of angiotensin II, histamine, prolactin, prostaglandins, and insulin-like growth factor (IGF) 1 [13,14,15,16].

The introduction of GnRH antagonists (GnRH-ant) to suppress the LH surge in IVF cycles and the GnRH agonist (GnRH-a) triggering followed by a “freeze all” policy have dramatically reduced if not eliminated the risk of OHSS [17]. However, the clinical and scientific interest in interventions aimed at preventing and treating OHSS is still high. This reflects that many IVF cycles worldwide are still performed employing the long GnRH-a down-regulation protocol. Furthermore, although the European Society of Human Reproduction and Embryology (ESHRE) guidelines suggest using GnRH-ant for IVF cycles in presumed high-risk patients [17], OHSS may still occur in presumed normal responders. Because reproductive outcome seems to be better for fresh embryo transfer in patients without polycystic ovary syndrome (PCOS) [18,19], this may further influence the choice to avoid elective embryo cryopreservation in clinical practice. Similarly, the utilization of a dual trigger and/or intensive luteal phase support, including the coadministration of GnRH-a plus low doses of hCG [20], may also increase the risk of OHSS [21]. Finally, new gonadotropin formulations have been studied in GnRH-a down-regulated IVF cycles [22,23,24], suggesting a new interest in the use of GnRH-a protocols. Collectively, these clinical strategies all suggest that OHSS is far from banished from clinical practice, and novel interventions will still be required.

Recently, we performed a systematic umbrella review [25] in accordance with the Preferred Reporting Items for Overviews of Reviews (PRIOR) guidelines [26] with the aim of identifying the best evidence-based interventions to prevent or reduce the incidence and severity of OHSS in patients undergoing IVF. A total of 33 interventions (used in 37 different clinical situations) were analyzed in 28 systematic reviews of RCTs with meta-analyses. Even if the quality assessment of the included studies was high-to-moderate for twenty-five studies, the certainty of evidence (CoE) was seen to be high-to-moderate only for six interventions [25]. Our analysis confirmed that the use of GnRH-ant should be preferred in presumed high-risk IVF patients and GnRH-a triggering with embryo freezing should be mandatory in case of persistent high-risk at the end of ovarian stimulation [25]. The progestin-primed ovarian stimulation (PPOS) protocol was shown to be a valid option in the case of elective embryo transfer, cancer patients in the context of fertility preservation, and donor patients [25]. In patients who undergo GnRH-a down-regulation, the use of mild stimulation was shown to be a safe approach, as well as metformin treatment during ovarian stimulation and dopamine agonists administration after ovulation triggering [25].

Moreover, the specific analysis of systematic reviews of RCTs with meta-analyses in that umbrella review is not only a strength, but also a limitation of the study because other interesting and potentially useful interventions were not included in the final analysis. Based on these considerations, the aim of the present study was to systematically review and discuss all interventions for the prevention of and reduction in the incidence and severity of OHSS in IVF patients not supported by systematic reviews of RCTs with meta-analyses, assess their efficacy, and grade them according to a well-validated tool for grading clinical evidence.

## 2. Methods

The protocol of the current review was registered on the PROSPERO website (Protocol study registration: PROSPERO CRD 268626, available at http://www.crd.york.ac.uk/PROSPERO, accessed 12 September 2023) and follows the PRISMA 2020 statement [27] (http://www.prisma-statement.org, accessed 12 September 2023) and the Population, Intervention, Comparison, Outcome (PICO) model [28]. No formal ethical approval was required because the study did not involve humans and/or the use of human tissue and/or hospital records samples, and no personal data were recorded and analyzed.

According to the PICO model [28], “Population” included women who undergo IVF/ICSI treatment, “Intervention” was considered each strategy used to reduce the risk and the severity of OHSS, “Comparison” included none or another strategy or placebo arm, and “Outcome” was considered the primary or secondary outcome of safety and efficacy, and its importance was classified to assess the effect of any intervention (https://gdt.gradepro.org/app/handbook/handbook.html, accessed 12 September 2023).

### 2.1. Literature Search

The search was initially performed, using the keywords “OVARIAN HYPERSTIMULATION SYNDROME” or “OHSS” in the following electronic databases: PubMed, The Cochrane Library, Web of Science, the World Health Organization (WHO) International Trials registry platform, Current Controlled Trials, and ClinicalTrial.gov. All publications within the database were considered with no time limits, with the searches re-run prior to the final analysis. The first search was performed to identify all potential interventions used/proposed to prevent or reduce the incidence and severity of OHSS. For that search, only comparative/controlled studies in humans published in the English language were included, and no further specific inclusion and exclusion criteria were considered. Subsequently, a further search was performed in the same databases using each specific intervention previously identified as the keywords. For each intervention, we searched and selected the studies with the highest hierarchy of evidence, as defined by the CEBM (http://www.cebm.ox.ac.uk, accessed 12 September 2023). Systematic reviews with or without meta-analyses were considered, followed by RCTs, prospective non-randomized, observational (cohorts, case–control, or cross-sectional) studies, and, finally, case series, experimental/translational studies, and expert opinion were searched and included in the final analysis. Among intercepted studies with the same grade of evidence, we included the most recent paper with the lowest risk of bias. Overlapping studies were included only if they had similar quality and were published in the same year or if the selected study did not report data on the endpoints considered. Preclinical/experimental studies, sub-analysis, non-comparative studies (not controlled for intervention, placebo, or other interventions), and network meta-analyses [29,30] were excluded from the final analysis. The authors also hand-searched the reference lists of the included articles and previous reviews to find additional data of interest for the aim of the present study.

All searches were performed by two authors (FC and AB) and checked by a third (DC). For each intervention, a specific table including the first author, year of publication, country, type of study, characteristics of the studied population, diagnostic criteria for OHSS, sample size, protocols used for ovarian stimulation, primary and secondary outcomes, quality of evidence (QoE, according to the risk of bias), and CoE was completed.

As this was a meta-analysis of published data, the original data were not as sought after by the authors.

### 2.2. Quality Assessment and Data Analysis

All studies included in the final analysis were analyzed for risk of bias, using specific tools according to the type of study. In particular, Assessing the Methodological Quality of Systematic Reviews 2 (AMSTAR-2) [31] (http://www.amstar.ca, accessed 12 September 2023), a Revised tool for Risk of Bias (rRoB 2) [32] (https://www.riskofbias.info/welcome/rob-2-0-tool, accessed 12 September 2023), Risk Of Bias in Non-randomized Studies–of Intervention (ROBINS-I) [33], and the Newcastle–Ottawa Scale (NOS) [34] were used, respectively, for systematic reviews, RCTs, prospective non-randomized studies, and observational/cohort studies. Case series were analyzed according to CAse REport (CARE) guidelines [35] (http://www.equator-network.org, accessed 12 September 2023). Concerning the rRoB 2 test, the risk of bias was reported in accordance with the tool as “high”, “low”, and “some concerns” [32].

For each intervention, alone or in combination, a qualitative analysis was performed using the data reported in the original manuscript. For all studies, the QoE was calculated after evaluating the risk of bias. In the case of meta-analyses, the CoE was reported as detailed in the original papers.

## 3. Results

The flowchart study according to the Preferred Reporting Items for Systematic Reviews and Meta-Analyses (PRISMA) 2020 guidelines [27] (http://www.prisma-statement.org, accessed 12 September 2023) is reported in Figure 1. A total of 48 interventions were identified, with 15 interventions not previously assessed by systematic review and meta-analysis of RCTs. The remaining 33 interventions were supported by type 1A according to the Oxford Centre for Evidence-Based Medicine (CEBM) evidence (Table 1) and have previously been discussed [25]. Of the fifteen interventions, two were not supported by available data, one (“hCG dose”) was supported by three different RCTs (one study for each specific clinical situation), ten interventions by ten RCTs, two interventions by systematic review without meta-analysis, and two interventions by two prospective studies (Table 1). Thus, a total of 15 studies were analyzed and discussed (Figure 1).

For each intervention identified, we provided the rationale for its use, the available/intercepted studies, the primary and secondary outcomes, the QoE, and the CoE (for systematic reviews). In Table 2, the characteristics of the studies included are detailed. Table 3 and Table 4 summarize the effects of each intervention in specific populations or clinical situations.

The quality assessment for RCTs and prospective studies is detailed in Figure 2 and in Table 5, respectively.

### 3.1. Use of Follitropin Delta for Ovarian Stimulation

Follitropin delta is a recombinant follicle-stimulating hormone (r-FSH) recently developed and expressed only in human retinal fetal cell lines [51]. Follitropin delta has a different pharmacokinetic profile from follitropin alpha due to the presence of a higher concentration of tri- and tetra-sialylated glycans and 2,6-linked sialic acid. This results in a more potent gonadotropin with the same follitropin delta dose in IU dose as follitropin alfa, leading to higher serum FSH concentrations, a greater follicular response, and higher estradiol concentrations [52].

Several RCTs have recently analyzed the efficacy and safety of follitropin delta [36,53,54,55,56,57,58]. The most recent study, with a low risk of bias, is a multi-center, assessor-blind RCT conducted on 1009 Asian patients randomized to receive follitropin delta or follitropin alpha in a GnRH-ant IVF protocol [36]. A significant reduction in the incidence of early OHSS and/or preventive interventions for early OHSS in patients stimulated with follitropin delta was observed in comparison to patients who received follitropin alpha (25/499 (5%) vs. 49/510 (9.6%), respectively; *p* = 0.004). Therefore, no significant difference in the incidence of moderate/severe OHSS (18/499 (3.6%) vs. 24/510 (4.7%) for follitropin delta vs. follitropin alpha, respectively; *p* = 0.365) was demonstrated [36]. Concerning reproductive outcomes, follitropin delta in comparison with follitropin alpha was associated with a higher live birth rate but not a different ongoing and clinical pregnancy rate, as well as a significantly lower number of oocytes retrieved [36]. The level of evidence was 1B (CEMB). The risk of bias was low (rRoB 2) [36].

### 3.2. FSH Dose Decrease for Ovarian Stimulation

Gonadotropin dose adjustment is commonly performed in clinical practice to optimize the safety and outcome during ovarian stimulation. Concerning the FSH dose decrease strategy, a systematic review without data synthesis [37], including 18 studies published from 2007 to 2017 for 6630 IVF cycles in which patients received a gonadotropin ovarian stimulation that allowed dose adjustment within the study protocol and that reported at least one dose adjustments of conventional r-FSH, concluded that decreasing the r-FSH dose during the mid-follicular phase of the ovarian stimulation may reduce the occurrence of OHSS compared to a fixed FSH dosage [37]. However, most trials evaluating the dose adjustment in predicted hyper-responders were designed to assess an individualized starting dose, confounding the available results. The QoE1 data were not applicable [37]. The level of evidence was 1C (CEMB). The QoE2 data were low (AMSTAR-2) [37].

### 3.3. Lower Doses of hCG for Ovulation Triggering

The role of hCG in OHSS pathogenesis has been widely demonstrated, and several molecular mechanisms are involved [7,8,9,10,11,12,13,14,15,16]. In particular, a close correlation between hCG concentration and VEGF mRNA has been demonstrated in patients who developed severe OHSS [59]. Thus, a dose decrease in hCG has been considered as a potential strategy to reduce OHSS risk [17]. At the moment, the most commonly used doses of hCG used in IVF until now has been 10,000 IU for u-hCG intramuscularly injected and 250 mcg for r-hCG subcutaneously administered.

#### 3.3.1. u-hCG

A double-blind RCT in an unselected population [38] demonstrated that a single dose of 5000 IU u-hCG compared to 10,000 IU u-hCG used for triggering resulted in a non-statistical difference in the incidence of OHSS (2% vs. 8.3%, *p* = 0.17). No difference between the two groups regarding oocytes retrieved and pregnancy rate was found [38]. The level of evidence was 1B (CEMB). The risk of bias reported some concerns (rRoB 2) [38].

A further RCT [39] in a selected high-risk population of 80 PCOS patients who received r-FSH for ovarian stimulation in GnRH-ant IVF cycles compared the administration of different hCG doses for triggering, i.e., 10,000 IU vs. 5000 IU vs. 2500 IU. A dose decrease in u-hCG to trigger final oocyte maturation did not affect the ongoing pregnancy rate, the early pregnancy loss, and the oocyte retrieved [39]. Concerning OHSS, only two cases of severe OHSS were reported (one patient in the 5000 UI group and one patient in the 10,000 UI group) [39]. The level of evidence was 1B (CEMB). The risk of bias reported some concerns (rRoB 2) [39].

#### 3.3.2. r-hCG

Concerning r-hCG, our search did not intercept comparative and/or controlled clinical studies on the use of lower r-hCG doses. An observational study [60] reported good reproductive outcomes and only one moderate OHSS case in thirty-five high-responder IVF patients who received 125 mcg r-hCG [60]. An RCT in a total of 180 patients compared 10,000 IU u-hCG (*n* = 60) to 500 μg (*n* = 60) and 250 μg (*n* = 60) r-hCG for ovulation triggering [40]. All the included patients underwent a GnRH-a long down-regulation protocol [40]. That study reported no statistical difference in OHSS incidence (6/60 (10%) vs. 4/60 (6.7%) vs. 3/60 (5%) for 10,000 IU u-hCG vs. 500 μg r-hCG vs. 250 μg r-hCG arms, respectively; *p* = 0.56). Regarding secondary outcomes, no difference was detected in clinical pregnancy rate or number of oocytes retrieved among groups [40]. The level of evidence was 1B (CEMB). The risk of bias reported some concerns (rRoB 2) [40].

### 3.4. Alternative Protocols for Ovulation Triggering

Different alternative protocols for ovulation triggering have been developed to improve the reproductive outcomes of IVF cycles, potentially also modifying OHSS risk.

#### 3.4.1. u-hCG Plus FSH for Ovulation Triggering

Even if the role of FSH surge for ovulation triggering is not completely understood in humans, experimental and animal data have demonstrated that it increases LH-receptor expression on granulosa cells, promotes the resumption of meiosis and cumulus expansion [61,62,63,64,65], and may trigger ovulation in the absence of LH activity [66,67,68]. Indirect human data from the use of the GnRH-a trigger have demonstrated a significant release of endogenous LH and FSH surges with potential positive effects on biological outcomes including higher oocyte recovery, maturity, and fertilization [69,70,71].

A recent double-blind, noninferiority RCT compared 1500 IU u-hCG plus 450 IU r-FSH (experimental) to 5000 IU or 10,000 IU u-hCG in 105 infertile patients scheduled for GnRH-a/GnRH-ant IVF cycles [41]. There was no OHSS case in the experimental group compared to two OHSS cases in the standard trigger groups. One patient had mild OHSS, whereas the other had severe OHSS requiring hospitalization for fluid management, anticoagulation, and paracentesis [41]. No difference was observed in live birth and miscarriage rates, but a slightly significant reduction in retrieved oocytes was found [41]. The level of evidence was 1B (CEMB). The risk of bias was low (rRoB 2) [41].

#### 3.4.2. GnRH-a Plus hCG vs. GnRH-a vs. hCG

Dual trigger is a strategy initially used as rescue treatment to improve implantation rates and the overall reproductive outcomes in IVF patients who received GnRH-a alone for ovulation triggering [72,73,74]. However, the co-administration of the hCG to GnRH-a trigger increases OHSS risk [75]. Recently, the use of GnRH-a plus hCG co-administration for final oocyte maturation was explored in patients with a normal ovarian response to improve both oocyte quality and reproductive outcomes compared to the hCG trigger alone [69,70], and significant efficacy of the dual triggering with GnRH-a plus hCG (vs. GnRH-a vs. hCG) has been also confirmed more recently in a large RCT, including 510 advanced-age IVF patients [71]. However, no direct data are available from IVF patients with PCOS and/or high OHSS risk currently.

A recent systematic review without data synthesis [42], including five retrospective studies, found no difference in OHSS risk between the use of dual triggering vs. GnRH-a triggering in four studies, whereas an increased risk of OHSS in patients who received dual triggering was observed in only one. Thus, the authors concluded that the incidence of OHSS was not significantly changed using dual triggering [42]. Insufficient evidence to support differences in live birth rate, clinical pregnancy, and miscarriage rates was found [42]. The QoE1 data were not applicable. The level of evidence was 1C (CEMB). The QoE2 data were low (AMSTAR-2) [42].

#### 3.4.3. Kisspeptin

Kisspeptin is a neuropeptide with a critical role in the function of the hypothalamic–pituitary–gonadal (HPG) axis, stimulating GnRH secretion from the hypothalamus and inducing gonadotropin secretion [76]. Only a few clinical trials have been published due to kisspeptin not currently being a licensed medication, limiting its use in clinical practice. Thus, only one phase 2 RCT in an IVF population of sixty women at high risk of OHSS explored the safety of kisspeptin-54 administration at different dosages [43]. Kisspeptin-54 was shown to be effective in triggering oocyte maturation without any moderate, severe, or critical OHSS event. In fact, in this study population, only 3/60 (5%) cases of mild early OHSS and 1/60 (2%) cases of mild late OHSS were reported [43]. The level of evidence was 1B (CEMB). The risk of bias reported some concerns (rRoB 2) [43].

### 3.5. Cycle Cancellation

In patients considered to be at a high risk of OHSS, cancellation of the cycles remains an option [17]. A cycle may be canceled before ovulation triggering in GnRH-a cycles (withholding hCG) or GnRH-ant cycles when elective cryopreservation is not possible. Our systematic research did not intercept specific and formal documents analyzing the efficacy of cycle cancellation as a strategy for preventing OHSS.

### 3.6. Elective Single Embryo Transfer (e-SET)

The risk and severity of OHSS are closely related to luteal hCG levels, which are significantly higher in multiple implantation pregnancies. However, direct data supporting the e-SET as a strategy to reduce OHSS risk are not available. Indirect evidence from a large prospective study [44] showed a close and significant association between the number of gestational sacs (±standard deviation) and the occurrence of late OHSS (1.67 ± 0.34) with early OHSS (0.36 ± 0.67) or no OHSS (0.37 ± 0.68) [44]. The level of evidence was 2B (CEMB). The QoE1 data were moderate (NOS) [44].

### 3.7. Aspirin

Aspirin inhibits the activity of the cyclooxygenase-1 enzyme, which results in a decrease in platelet activity and a reduction in the risk of blood clotting, altering the pathological cascade caused by VEGF [77]. Different RCTs were intercepted from the literature [45,78,79].

The most recent study is a double-blind placebo-controlled RCT [45] that demonstrated no difference in the incidence of moderate-to-severe OHSS between low-dose aspirin (100 mg daily) vs. the placebo in 214 infertile PCOS patients scheduled for GnRH-a IVF programs (38/109 (34.9%) vs. 32/105 (30.5), respectively; *p* = 0.494). No difference was found in the number of oocytes retrieved and the clinical pregnancy rate [45]. The level of evidence was 1B (CEMB). The risk of bias was low (rRoB 2) [45].

### 3.8. Ketoconazole

Ketoconazole is an inhibitor of the steroidogenic enzyme P450 in the adrenal cortex and gonads and is a potential modulator of the ovarian response to gonadotropin [80].

Two RCTs were found in our research [46,81]. The highest quality RCT, a double-blind, placebo-controlled study [46], showed that ketoconazole administration did not prevent OHSS (4/50 (7%) vs. 5/51 (9%), for ketoconazole vs. the placebo group, respectively; *p* > 0.05) in PCOS patients. No differences in pregnancy rates were detected between arms that received and did not receive ketoconazole [46]. The level of evidence was 1B (CEMB). The risk of bias was low (rRoB 2) [46].

### 3.9. Luteal GnRH-Ant Administration

The administration of GnRH-ant during the luteal phase was studied as a potential intervention to prevent early OHSS and reduce the severity of OHSS [82]. GnRH-ant injections suppress the release of LH by the pituitary, enhancing luteolysis and inducing a significant reduction in circulating VEGF [83].

A prospective study in a total of 105 patients at a high risk of OHSS concluded that GnRH-ant administration for three days was effective in reducing the moderate-to-severe OHSS incidence and inducing a faster regression of OHSS symptoms (11/61 (18.03%) vs. 13/35 (37.14%), *p* = 0.03) [47]. No data on reproductive outcomes are available and no difference in the hospital admission and average hospitalization duration were seen [47]. The level of evidence was 2B (CEMB). The quality of the data was high (NOS) [47].

### 3.10. hCG Administration for Intensified Luteal Phase Support

Small doses of hCG, as luteal phase supports, were tested in high responders who received ovulation triggering with GnRH-a in GnRH-ant IVF cycles [84,85,86,87]. A recent RCT [48] in 212 infertile IVF patients who received GnRH-a for triggering compared fresh transfer to a freeze-all policy. In the fresh transfer group, patients were administered a bolus of 1500 IU hCG on the day of oocyte retrieval in addition to oral estradiol and vaginal micronized progesterone for luteal phase support [48]. Moderate–severe OHSS occurred only in the low-dose hCG group (9/105 (8.9%) vs. 0/104 (0%); risk difference (RD) −8.6%, 95% CI −13.9% to −3.2, *p* < 0.01) [48]. No difference between the two groups was found in clinical pregnancy, live birth, miscarriage, or oocyte retrieval rates [48]. The level of evidence was 1B (CEMB). The risk of bias was low (rRoB 2) [48].

### 3.11. Inositol

Inositol is a compound of biological origin that is involved in numerous biological processes including cellular signaling [88]. Positive effects have been demonstrated with the use of inositol supplementation in women with PCOS [89]. Inositol efficacy on OHSS risk may be due to a beneficial effect on the abnormal ovarian/follicle dynamics [88,89] but also the expression of cyclooxygenase type 2 (COX-2) and VEGF with suppression of vascular permeability [90].

Several RCTs related to the efficacy of inositol were intercepted [49,91,92]. The most recent double-blind RCT [49] in a total of 102 infertile PCOS IVF patients compared the effect of 3-month myoinositol and metformin pretreatment. No statistically significant difference in OHSS incidence was found between the two groups (5/50 (10%) vs. 10/50 (20%) for myoinositol and metformin group, respectively; *p* = 0.10). A significantly higher clinical pregnancy rate was found in myoinositol-treated patients, whereas no difference was observed in terms of oocytes retrieved [49]. The level of evidence was 1B (CEMB). The risk of bias was low (rRoB 2) [49].

### 3.12. Insulin Sensitizing Drugs (ISDs)

All data on the effects of ISDs regarding OHSS risk involve the administration of metformin. No data on other ISDs, including rosiglitazone, pioglitazone, sulfonylurea, peroxisome proliferator-activated receptor agonists, liraglutide, semaglutide, glucagon, α-glucosidase inhibitors, and sodium–glucose cotransporter (SGLT)-2 inhibitors were intercepted about the effect on OHSS risk in IVF patients. The effect of metformin on the risk of OHSS has recently been reported elsewhere [25].

### 3.13. Vitamin D

Many experimental data seem to suggest a role of vitamin D in the female reproductive system. Vitamin D supplementation may improve ovulatory function and oocyte and embryo quality, especially in vitamin D-deficient patients [93]. A significant inverse relationship was detected between ovarian reserve and serum vitamin D levels [93,94]. Vitamin D supplementation in women with PCOS reduced the serum AMH levels, whereas in ovulatory women without PCOS, a significant AMH increase was observed [94]. The potential effect of vitamin D on serum VEGF levels has been also studied with controversial results. Animal studies showed that vitamin D administration did not exert a direct effect on VEGF and OHSS development in non-vitamin D-deficient female Wistar rats [95], whereas a beneficial effect of vitamin D supplementation on VEGF levels and other inflammatory patterns was detected in vitamin D-deficient women with PCOS [96,97,98].

Many systematic reviews with [99,100,101,102,103,104,105] or without [106] meta-analyses about the effect of vitamin D levels and/or vitamin D supplementation in IVF patients were identified. However, all these studies analyzed and discussed many reproductive outcomes with quite different conclusions and did not provide results on the effect of vitamin D on OHSS risk. The analysis of the different RCTs available [50,107,108,109,110,111,112,113] showed that only one RCT included as a secondary endpoint the adverse events, including high OHSS risk [50]. In particular, in a double-blind placebo-controlled trial [50], a total of 630 infertile IVF patients with low vitamin D levels (less than 30 ng/mL) were randomized to receive 600,000 IU vitamin D pretreatment (from 2 to 12 weeks before IVF) or a placebo. No statistically significant difference in patients who did not receive the embryo transfer for a high risk of OHSS (75/285 (26.3%) vs. 76/288 (26.4%) for the vitamin D vs. placebo group, respectively) was observed [50]. No difference between the vitamin D and placebo group was detected in any of the reproductive outcomes assessed [50]. The level of evidence was 1B (CEMB). The risk of bias was low (rRoB 2) [50].

## 4. Discussion

The current study aimed to identify interventions to prevent or reduce OHSS risk with clinical evidence lower than type 1A, according to the CEBM hierarchy of evidence. In fact, in a recent umbrella review [25], only systematic reviews of RCTs with meta-analyses were included, and many interesting and potentially useful treatment regimens were not considered. In the present paper, 15 further interventions were identified and discussed evaluating the risk/benefit ratio. Each intervention was analyzed considering the best evidence available according to the CEBM hierarchy (highest), the risk of bias (lowest), and the year of publication (more recent). Although we used the CEBM system to grade the evidence, which has not been updated since 2009, it represents a simple and reproducible tool to evaluate the available scientific literature (http://www.cebm.ox.ac.uk, accessed 12 September 2023). In this regard, many interventions for minimizing OHSS risk were supported by systematic reviews without meta-analyses or RCTs.

Our findings suggest that follitropin delta in GnRH-ant IVF cycles may be efficacious in terms of the reduction in early OHSS risk without a negative effect on reproductive outcomes and improve the live birth rate [36]. These results on OHSS risk are in accordance with two large multicenter RCTs [54,56]. The first was an assessor-blinded, noninferiority RCT that demonstrated a reduction in moderate-to-severe OHSS incidence and/or its preventive interventions in 1329 Caucasian patients with polycystic ovaries using an individualized follitropin delta administration in comparison to conventional follitropin alpha [54]. The second was another assessor-blind, noninferiority RCT trial performed on 347 Japanese IVF patients that reported an overall risk of OHSS (early and late OHSS), and a moderate–severe OHSS risk was reduced to approximately half with the use of individualized follitropin delta administration in comparison to the standard regimen [56]. Of note, the use of an individualized follitropin delta dosing regimen achieves significantly and clinically higher live birth rates compared to the women who received a conventional follitropin alfa regimen with a relative increase of more than 25% [36], and these findings are partially in agreement with other RCTs that showed no clinically significant difference in reproductive outcomes between two protocols [54] and a better live birth rate per started cycle with an individualized follitropin delta protocol [56]. A secondary analysis of two clinical trials, designed to assess OHSS risk in 1326 patients who received sequential ovarian stimulation cycles with follitropin delta, demonstrated that follitropin delta, administered in individualized dosing in comparison to a conventional follitropin alfa protocol, significantly reduced the risk of moderate-to-severe OHSS and preventive interventions [55]. The greatest benefit was observed in patients in the highest anti-Müllerian hormone quartile. Unfortunately, controlled data on the safety and efficacy of follitropin delta in GnRH-a IVF cycles and presumed hyper-responder patients are needed [114].

Surprisingly, only a few confounded data are available about the use of gonadotropin dose decrease, a strategy well-supported by common sense and used in up to 41% of IVF cycles in the United States [115]. On the other hand, much data have been published about the tailoring/personalization of the starting dose of gonadotropins during the last years with single or multiple parameters and are also combined in specific algorithms [25], and the dose adjustment of the initial dose of gonadotropin is frequently included in the conventional arm (vs. the individualized arm) [37]. In the systematic review by Fatemi et al. [37], only three studies reported direct comparisons of outcomes between constant dose vs. dose adjustment groups [54,116,117], and the many confounders and biases did not permit solid conclusions about the incidence of dose adjustment in routine clinical practice and its impact on clinical outcomes.

International guidelines [17] suggest the use of low doses of hCG for triggering ovulation in case of high OHSS risk, and this is particularly true for GnRH-a IVF cycles. Moreover, our analysis revealed a reduction in OHSS incidence in high-risk patients only with the low doses of u-hCG administrated in GnRH-ant cycle. However, it is interesting to note that the lower doses of r-hCG (250µg) are widely used in clinical practice as standard treatment but the larger 10,000 IU dose of u-hCG is still commonly administrated. Interestingly, data regard the use of very low u-hCG dose plus high-dose r-FSH bolus in IVF patients not at a high risk for OHSS, even if the presumed efficacy on OHSS incidence needs to be confirmed in other settings on larger study samples and in high-risk populations. Efficacy and safety data on kisspeptin, and its analogs, as ovulation triggers, are still limited to experimental settings. On the other hand, inositol pretreatment and GnRH-ant administration during the luteal phase may reduce the severity and duration of OHSS. Finally, our data, in agreement with a recent network meta-analysis [118], does not support the use of aspirin and ketoconazole for OHSS prevention in IVF.

Our findings underline that vitamin D supplementation is not effective as a preventive measure for reducing OHSS risk, although it is advised prior to pregnancy for all women [119]. A large multicenter randomized, double-blinded, placebo-controlled trial, evaluating the effectiveness of oral capsules of 4000 IU vitamin D per day given as a pretreatment and cotreatment (from 12 weeks before up to the day of triggering) in IVF patients with PCOS is in progress [120]. The incidence of moderate and severe OHSS will be reported as the secondary outcome [120], and the results are awaited with interest.

We decided to exclude systematic reviews with network meta-analyses from our protocol design because their results are still under debate, considering the unclear assumption about transitivity among comparisons [29,30,121]. However, two recent network meta-analyses of RCTs [118,122] have been recently published and concluded that algorithm-based strategies are more effective in reducing OHSS compared to experience-based treatment and standard gonadotropin dosing [122], and HES and cabergoline are the only treatments for reducing the incidence of OHSS compared to a placebo or blank controls [118].

The present review has several strengths. These include the extensive literature search of specific potential interventions with an impact on OHSS, the use of the PICO model [28], and the performance of a careful quality assessment for each intervention with specific tools according to the study design. We do, however, acknowledge several limitations, including the low quality of several studies included in the analysis. In addition, almost all studies discussed were not designed and powered to detect differences in OHSS incidence, data on maternal mortality and morbidity were poorly reported, and many studies did not even highlight differences in terms of live births. In many studies, the risk of OHSS was reported as a secondary outcome and was not tested in populations at a high risk of OHSS. For example, in Anaya’s study, the IVF patients considered to be at the highest risk for OHSS, i.e., serum estradiol levels higher than 5000 pg/mL on the day of the trigger, were excluded from randomization for safety concerns, limiting the clinical application of the trial findings to at-risk patients [41]. Finally, another limitation may be due to the use of rRob2 to evaluate the RCTs; we feel that this tool is not sensible enough to discriminate the best study quality considering that it includes only three possible categories. This may have introduced a significant confounder due to an incomplete interception of more relevant data.

## 5. Conclusions

The present systematic review identified several treatments/strategies that are potentially effective in reducing the incidence and severity of OHSS, even if not supported by the highest clinical evidence. Many clinical data regard interventions not based on well-defined experimental studies. Further basic research is certainly needed to clarify the molecular mechanisms involved in the pathogenesis of OHSS to identify the crucial therapeutic targets.

## Figures and Tables

**Figure 1 ijms-24-14185-f001:**
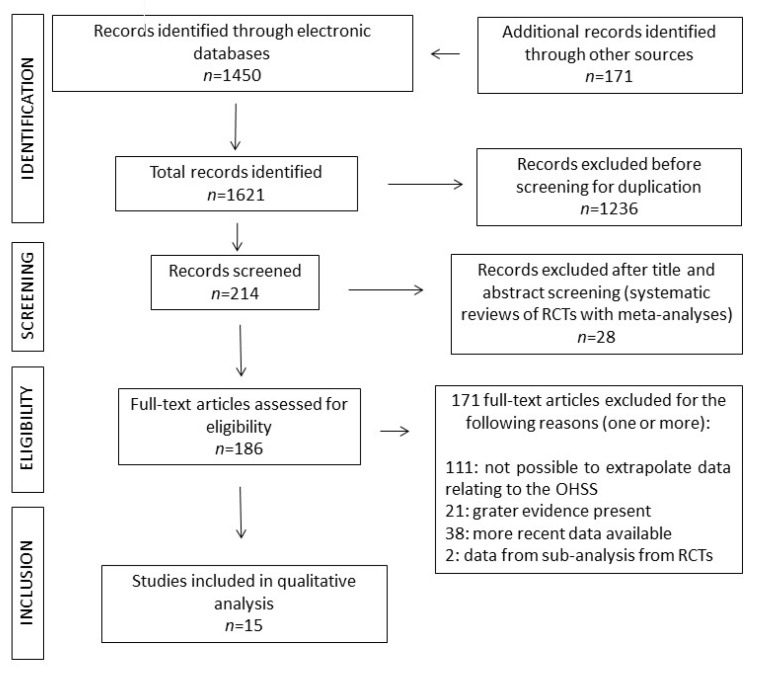
PRISMA flowchart [27].

**Figure 2 ijms-24-14185-f002:**
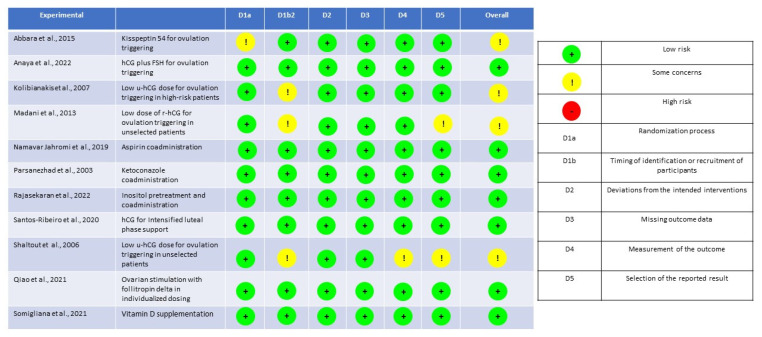
Assessment of the risk of bias for RCTs using the Revised tool for Risk of Bias (rRoB 2) [32]. (Abbara et al., 2015 [43]; Anaya et al., 2022 [41]; Kolibianakis et al., 2007 [39]; Madani, et al., 2013 [40]; Namavar Jahromi et al., 2019 [45]; Parsanezhad et al., 2003 [46]; Rajasekaran et al., 2022 [49]; Santos-Ribeiro et al., 2020 [48]; Shaltout al., 2006 [38]; Qiao et al., 2021 [36]; Somigliana et al., 2021 [50]).

**Table 1 ijms-24-14185-t001:** All interventions identified to potentially modify OHSS risk are classified according to according to the CEBM recommendations (http://www.cebm.ox.ac.uk, accessed 12 September 2023). Interventions graded as 1A have been discussed elsewhere [25].

Interventions	Scientific Evidence
**Alternative hCG protocol**	1B
**Aspirin**	1B
**Calcium infusion**	1A
**Cabergoline**	1A
**Clomiphene citrate**	1A
**Cycle cancellation**	NA
**Coasting**	1A
**Corifollitropin alfa**	1A
**Diosmin**	1A
**Dopaminergic agonists**	1A
**Dual trigger**	1C
**Elective cryopreservation**	1A
**Elective single embryo transfer**	2B
**Follitropin delta**	1B
**FSH dose decrease**	1C
**Glucocorticoid**	1A
**GnRH analogs**	1A
**Inositol**	1B
**Insulin sensitizing drugs**	NA
**In vitro maturation of oocytes**	1A
**Intensified luteal phase support with hCG**	1B
**Intensified luteal phase support: GnRH agonist**	1A
**Ketoconazole**	1B
**Kisspeptin**	1B
**Letrozole**	1A
**LH addition**	1A
**Luteal GnRH antagonist administration**	2B
**Luteal phase support/GnRH agonist**	1A
**Luteal phase support/hCG**	1A
**Luteal phase support/progesterone**	1A
**Melatonin**	1A
**Metformin**	1A
**Mild ovarian stimulation**	1A
**Monitoring and surveillance**	1A
**Natural IVF cycles**	1A
**Oral contraceptives**	1A
**Ovarian drilling**	1A
**Personalization**	1A
**Predictive models**	1A
**Progestin-primed ovarian stimulation**	1A
**Triggering/GnRH agonist**	1A
**Triggering/r-hLH**	1A
**Triggering/hCG dose**	1B
**Triggering/hCG type**	1A
**Gonadotropins**	1A
**Vitamin D**	1B
**Volume expanders/albumin**	1A
**Volume expanders/hydroxyethyl starch**	1A

FSH: follicle-stimulating hormone; GnRH: gonadotropin-releasing hormone; hCG: human chorionic gonadotropin; IVF: in vitro fertilization; LH: luteinizing hormone; NA: not available data; r-hLH: recombinant human.

**Table 2 ijms-24-14185-t002:** Characteristics of the studies included in the final analysis according to the specific intervention.

Interventions	Evidence	Country	Type of Study	Ovarian Stimulation Protocol	Population	Risk of Bias	CoE
**Follitropin delta**	Qiao et al., 2021 [36]	China	RCT	1009 Asian patients randomized 1:1 to follitropin delta dose based on AMH and body weight or conventional dosing with follitropin alfa following a GnRH-ant protocol.	Presumed normo-responder infertile population	Low ^a^	/
**FSH dose decrease**	Fatemi et al., 2021 [37]	Emirates	SR	Included studies from 2007 to 2017 on women receiving ART treatment that allowed dose adjustment within the study protocol and that reported ≥ 1 dose adjustments of r-FSH.	General infertile population	Not applicable	Low ^b^
**Dose of hCG for ovulation triggering: u-hCG in unselected population**	Shaltout al., 2006 [38]	Egypt	RCT	98 infertile patients randomized 1:1 to 5000 IU or 10,000 IU u-hCG for ovulation induction.	General infertile population	Some concerns ^a^	/
**Dose of hCG for ovulation triggering: u-hCG in high-risk population**	Kolibianakis et al., 2007 [39]	Belgium	RCT	75 infertile patients randomized 1:1:1 to 10,000 IU, 5000 IU or 2500 IU of u-hCG for ovulation induction in GnRH-ant cycles.	PCOS patients	Some concerns ^a^	/
**Dose of hCG for ovulation triggering: r-hCG in presumed normo-responders**	Madani, et al., 2013 [40]	Iran	RCT	180 infertile patients randomized 1:1:1 to 10,000 IU u-hCG or 250 μg r- hCG or 500 μg r-hCG for ovulation induction in GnRH-a long protocol.	Presumed normal-risk population	Some concerns ^a^	/
**Alternative protocols for ovulation triggering: FSH *plus* u-hCG**	Anaya et al., 2022 [41]	US	RCT	105 patients undergoing IVF were randomized to receive an alternative trigger of 1500 IU of u-hCG plus 450 IU of FSH or a standard trigger dose of u-hCG (5000 or 10,000 IU) for final oocyte maturation.	General infertile population	Low ^a^	/
**Alternative protocols for ovulation triggering: GnRH-a *plus* hCG**	Vyrides et al., 2022 [42]	UK	SR	5 retrospective studies were included. Dual trigger with GnRH-a were compared for final oocyte maturation in high responders undergoing GnRH-ant cycles.	High responders’ patients	Not applicable	Low ^b^
**Kisspeptin**	Abbara et al., 2015 [43]	UK	RCT	60 women at high risk of developing OHSS after a standard rFSH/GnRH-ant protocol were randomized to receive a single injection of kisspeptin-54 to trigger oocyte maturation using an adaptive design for dose allocation.	High risk OHSS patients	Some concerns ^a^	/
**Elective single embryo transfer**	Mathur et al., 2000 [44]	UK	Prospective study	Comparison of patient and cycle characteristics among three study groups: early OHSS (n = 2284), late OHSS (n = 48), and non-OHSS (n = 30).	General infertile population	/	High ^c^
**Aspirin**	Namavar Jahromi et al., 2019 [45]	Iran	RCT	232 infertile PCOS patients randomized 1:1 to low-dose of aspirin or placebo.	PCOS patients	Low ^a^	/
**Ketoconazole**	Parsanezhad et al., 2003 [46]	Germany	RCT	101 PCOS patients randomized 1:1 to ketoconazole (50 mg every 48 h) or placebo (every 48 h).	PCOS patients	Low ^a^	/
**Luteal GnRH antagonist administration**	Zeng et al., 2019 [47]	China	Prospective cohort study	105 patients with high-risk OHSS undergoing cryopreservation of all embryos were randomized 1:1 to luteal GnRH-ant (0.25 mg cetrorelix daily from days 3 to 5 POR) or no drug/intervention.	High risk OHSS patients	/	High ^c^
**hCG for intensified luteal phase support**	Santos-Ribeiro et al., 2020 [48]	Portugal	RCT	212 patients following GnRH- a triggering randomized 1:1 to freeze-all approach or fresh embryo transfer using a low-dose of hCG for intensified luteal phase support.	Women with an excessive response to ovarian stimulation	Low ^a^	/
**Inositol**	Rajasekaran et al., 2022 [49]	India	RCT	102 infertile PCOS patients randomized 1:1 to 2 g myoinositol twice daily or 850 mg metformin twice daily.	PCOS patients	Low ^a^	/
**Vitamin D**	Somigliana et al., 2021 [50]	Italy	RCT	630 infertile patients randomized 1:1 to receive 600,000 IU vitamin D pretreatment or placebo from 2 to 12 weeks before IVF.	Good responder patients without contraindication to vitamin D	Low ^a^	/

Data on cycle cancellation and insulin-sensitizing drugs are not reported because no direct clinical data are available. AMH: anti-Müllerian hormone; ART: assisted reproductive technology CoE: certainty of evidence GnRH: gonadotropin-releasing hormone; GnRH-a: GnRH-agonist; GnRH-ant: GnRH-antagonist; hMG: human menopausal gonadotrophin; IU: international unit; IVF: in vitro fertilization; FSH: follicle-stimulating hormone; OHSS: ovarian hyperstimulation syndrome; PCOS: polycystic ovary syndrome; POR: post-oocyte retrieval; r-FSH: recombinant-human FSH; RCT: randomized controlled trial; SR: systematic review; UK: United Kingdom; u-hCG: urinary human chorionic gonadotropin; US: United States. Legend: ^a^ Revised tool for Risk of Bias (rRoB 2) [32]; ^b^ Assessing the Methodological Quality of Systematic Reviews 2 (AMSTAR-2) [31]; ^c^ Newcastle–Ottawa Scale (NOS) [34].

**Table 3 ijms-24-14185-t003:** Primary and secondary endpoints for each specific intervention. Only quantitative data are reported.

	Total OHSS	Moderate-Severe OHSS	Live Births	Clinical Pregnancies	Ongoing Pregnancies	Pregnancies	Miscarriages	Oocytes Retrieved	Hospital Admission	Hospitalization Duration
**Follitropin delta [36] ^a^**	**Reduction ^c^** 25/499 (5%) vs. 49/510 (9.6%); *p* = 0.004	**No difference** 18/499 (3.6%) vs. 24/510 (4.7%); *p* = 0.365	**Increased** 156/499 (31.3%) vs. 126/510 (24.7%); *p =* 0.023	**No difference** 180/499 (36.1%) vs. 159/510 (31.2%); *p* = 0.099	**No difference** 156/499 (31.3%) vs. 131/510 (25.7%); *p =* 0.058	/	/	**Reduction** 10.0 ± 6.1 vs. 12.4 ± 7.3; *p <* 0.001	/	/
**Low hCG dose for ovulation triggering: u-hCG, unselected patients [38] ^a^**	**No difference** 1/50 (2%) vs. 4/48 (8.3); *p =* 0.17	/	**/**	**/**	**/**	**No difference** 17/50 (33.3%) vs. 17/48 (35.4%); *p =* 0.75	/	**No difference** 7 ± 3.5 vs. 7.4 ± 3; *p =* 0.54	/	/
**Low hCG dose for ovulation triggering: u-hCG, high risk patients [39] ^a^**	/	**No difference** 1/28 (3.6%) vs. 1/26 (3.8%) vs. 0/26 (0%); *p =* NA *	**/**	**/**	**No difference** 25 (12.7–43.4) (7/28) vs. 30.8 (16.5–49.9) (8/26); *p =* 0.64	/	**No difference** 30 (10.8–60.3) vs. 27.3 (9.7–56.5); *p =* 0.89	**No difference** 14 (9.0) vs. 11.5 (10.0) vs. 9 (7.0); *p =* 0.35	/	/
**Low hCG dose for ovulation triggering: r-hCG, normo-responders [40] ^a^**	**No difference** 6/60 (10%) vs. 4/60 (6.7%) vs. 3/60 (5%); *p =* 0.56	/	**/**	**No difference** 19/55 (34.5%) vs. 19/45 (42.2%) vs. 23/53 (43.4%); *p =* 0.60	**/**	**/**	**/**	**No difference** 12.25 ± 5.30 vs. 12.40 ± 6.44 vs. 11.37 ± 5.3; *p =* 0.56	/	/
**Alternative protocols for ovulation triggering: FSH *plus* u-hCG [41] ^a^**	**Reduction** 0/54 (0%) vs. 2/51 (3.9%); *p* = NA	/	**No difference** 26/54 (48.1%) vs. 32/51 (62.7%) RR 0.73, 95%CI 0.48, 1.11	**/**	**/**	/	**No difference** 3/27 (11.1%) vs. 3/33 (9.1%)	**Reduction** 13.4 (0.85) vs. 16.1 (1.03); WM 0.83, 95%CI 0.70, 0.995; *p* = 0.045	/	/
**Kisspeptin [43]**	4/60 (7%) cases of mild OHSS	**No cases**	**/**	**/**	**/**	/	/	**/**	/	/
**Elective single embryo transfer [44] ^b^**	**Increased in multiple pregnancies** Late OHSS (1.67 ± 0.34) vs. late OHSS (0.36 ± 0.67) vs. no-OHSS (0.37 ± 0.68); *p <* 0.001	/	**/**	**/**	**/**	/	/	**/**	/	/
**Aspirin [45] ^a^**	**/**	**No difference** 38/109 (34.9%) vs. 32/105 (30.5); *p =* 0.494	**/**	**No difference** 31/109 (28.4%) vs. 24/105 (22.9); *p =* 0.350	**/**	/	/	**No difference** 10.92 ± 6.27 vs. 10.73 ± 6.05; *p =* 0.819	/	/
**Ketoconazole [46] ^a^**	**No difference** 4/50 (7%) vs. 5/51 (9%); *p* > 0.05	/	**/**	**/**	**/**	**No difference** 9/50 (18%) vs. 11/51 (21.5%); *p* > 0.05	/	/		
**Luteal GnRH antagonist administration [47] ^b^**	**/**	**Reduction** 11/61 (18.03%) vs.13/35 (37.14%); *p =* 0.03	**/**	**/**	**/**	/	/	**/**	**No difference** 4/61 (6.58%) vs. 7/35 (20%); *p =* 0.073	**No difference** 5.75 ± 0.96 vs.8.57 ± 1.90; *p =* 0.03
**hCG for intensified luteal phase support [48] ^a^**	**/**	**Increased** 9/105 (8.9%) vs. 0/104 (0%); RD −8.6%, 95% CI −13.9%, - 3.2; *p <* 0.01	**No difference** 41/104 (48.6%) vs. 42/101 (54.8%); RD 6.2%; 95% CI −7.3, 19.8; *p =* 0.41	**No difference** 51/105 (48.6%) vs. 57/104 (54.8%); RD 6.2%; 95% CI −7.3, 19.8; *p =* 0.41	**/**	/	**No difference** 9/51(17.6%) vs. 12/57(21%); RD 3.4%; 95% CI −11.5, 18.3; *p =* 0.81	**No difference** 18.5 ± 7.1 vs. 19.4 ± 7.8; RD −0.9; 95% CI −1.2, 2.9; *p =* 0.66	/	/
**Inositol [49]**	**No difference** 5/50 (10%) vs. 10/50 (20%); *p =* 0.10	**/**	**/**	**/**	**/**	**Increased** 18/50 (36%) vs. 9/50 (18%); *p =* 0.09	**/**	**No difference** 14 (0–18) vs. 12 (0–16); *p =* 0.13	/	/
**Vitamin D [50]**	**No difference** 75/285 (26.3%) vs. 76/288 (26.4%); *p >* 0.05	**/**	**No difference** 98/308 (32%) vs. 110/322 (34%); *p* = 0.55	**No difference** 113/285 (37%) vs. 130/288 (40%); RR 0.91, 95% CI 0.75, 1.11; *p* = 0.37	**/**	**No difference** 54/169 (32%) vs. 52/159 (33%); *p* = 0.91	**No difference** 13/113 (12%) vs. 16/130 (12%); *p* = 0.85	**No difference** 6 (4–9) vs. 6 (3–9); *p* = 0.13	**/**	**/**

CI: confidence interval; hCG: human chorionic gonadotropin; GnRH: gonadotropin-releasing hormone; NA: not applicable; OHSS: ovarian hyperstimulation syndrome; OR: odds ratio; NA: not available data; RD: risk difference; r-hCG: recombinant hCG; RR: relative risk; u-hCG: urinary hCG. Legend: ^a^ randomized controlled trial; ^b^ prospective study; ^c^ reduction in the incidence of early OHSS and/or preventive interventions for early OHSS; * calculated because not reported in the main document.

**Table 4 ijms-24-14185-t004:** Results related to systematic review without meta-analysis of primary and secondary endpoints for each specific intervention.

Intervention	Included Studies (*n*)	OHSS Incidence	Live Births	Clinical Pregnancies	Ongoing Pregnancies	Pregnancies	Miscarriages	Oocytes Retrieved
**FSH dose decrease [37]**	18	Unclear	No difference	Unclear	/	Unclear	/	/
**Dual trigger [42]**	5	Unclear	Unclear	Unclear	Unclear	/	No difference	/

**Table 5 ijms-24-14185-t005:** Risk of bias for prospective studies performed by Newcastle-Ottawa Scale (NOS) [34].

Author (Year)	Selection	Comparability	Outcome	Total NOS Score	Quality
**Mathur et al., 2000 [44]**	+++	+	++	6	Moderate
**Zeng et al., 2019 [47]**	++++	++	++	8	High

Legend: NOS evaluates the risk of bias in three different areas: 1. group selection (including representativeness of the exposed cohort, selection of non-exposed, ascertainment of exposure, and outcome not present at start); 2. group comparability; and 3. determination of exposure and outcome (including evaluation of outcome, adequate follow-up length, and adequacy of follow-up). For each specific area, a maximum of one point for each item within the “Selection” and “Exposure” categories, and a maximum of two points for “Comparability” category can be given. A total score ranging from 0 to 9 can be attributed. Based on these standards, studies are classified with low (score between 9 and 7), moderate (score between 6 and 4) and high (score between 3 and 0) risk of bias.

## Data Availability

The data underlying this article will be shared upon reasonable request to the corresponding author.

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
