# Peer review of "Beyond the Umbrella: A Systematic Review of the Interventions for the Prevention of and Reduction in the Incidence and Severity of Ovarian Hyperstimulation Syndrome in Patients Who Undergo In Vitro Fertilization Treatments"

_ijms, 2023, doi:10.3390/ijms241814185_

Round 1

Reviewer 1 Report

This narrative review of the various ideas and trials to prevent/treat OHSS has some value for those interested in the history of empirical treatments. Otherwise the aim of this review and its additive value to the potential readers is not clear. Naturally in the early years of COH the molecular explanation for development of OHSS was not fully understood, so many ideas were suggested and tried. Presently situation has changed and the possibility to induce ovulation/final oocyte maturation by the use of GnRH agonist to induce endogenous LH surge has revolutionized the prevention of OHSS ans became a standard. As no other intervention was proven challenging this concept the manuscript did not explain the scientific value of  the manuscript and how it may add to the knowledgebase of those interested in the field of controlled ovarian hyperstimulation. 

If accepted the manuscript may be improved by some shortening.

Author Response

Manuscript ID: ijms-2567477
Type of manuscript: Review
Title: Beyond the umbrella: a systematic review of the interventions for prevention of and reduction in the incidence and severity of ovarian hyperstimulation syndrome in patients who undergo in vitro fertilization
treatments.
Authors: Stefano Palomba *, Flavia Costanzi, Scott M. M Nelson, Aris Besharat, Donatella Caserta, Peter Humaidan

Reviewer

This narrative review of the various ideas and trials to prevent/treat OHSS has some value for those interested in the history of empirical treatments. Otherwise the aim of this review and its additive value to the potential readers is not clear. Naturally in the early years of COH the molecular explanation for development of OHSS was not fully understood, so many ideas were suggested and tried. Presently situation has changed and the possibility to induce ovulation/final oocyte maturation by the use of GnRH agonist to induce endogenous LH surge has revolutionized the prevention of OHSS and became a standard. As no other intervention was proven challenging this concept the manuscript did not explain the scientific value of  the manuscript and how it may add to the knowledgebase of those interested in the field of controlled ovarian hyperstimulation.

Authors

We thank the Reviewer for the opportunity to revise the manuscript and have developed a point-by-point response below. We hope that the revised manuscript is now suitable for publication.

The primary purpose of the manuscript is to highlight additional OHSS preventative strategies that are potentially less well known, not been subjected to systematic review, and hence not included in a recent umbrella review. As the PRIOR guidelines for umbrella reviews excludes some potentially effective treatments as they have not been subject to type A evidence (systematic review), rit is appropriate that clinicians are aware of the totality of potential interventions and the strength of the evidence underlying them. Our paper also highlights several treatments which should be not used in the clinical practice because they are not supported by strong evidence. We have revised the manuscript to make the overall aim clearer.

Reviewer 2 Report

Ovarian hyperstimulation syndrome (OHSS) is a serious medical condition, often due to the controlled ovarian stimulation during assisted reproductive technologies (ART) therapy. According to the available data, moderate to severe OHSS is observed in 1- 5% of the cycles performed. Therefore, the manuscript treats a contemporary issue, and can be beneficial predominantly to ART practitioners. The authors present data for the effectiveness of the different approaches for prevention of and reduction in the incidence and severity of OHSS.

I accept the presented manuscript as a continuation of the beforehand published paper in 2023 of the same authors: (Palomba, S., Costanzi, F., Nelson, S.M. et al. Interventions to prevent or reduce the incidence and severity of ovarian hyperstimulation syndrome: a systematic umbrella review of the best clinical evidence. Reprod Biol Endocrinol 21, 67 (2023). https://doi.org/10.1186/s12958-023-01113-6 ).

The methodology of the review and the statistical analysis are both well described. However, I have the following recommendations to the manuscript:

1.      As a whole the work is based on statistical analysis of existing data, there is too little to no molecular biology and I believe that the manuscript will be more appropriate to be published in clinical or endocrinological journal. Therefore, I think that in the introduction more information can be added regarding the molecular mechanisms for OHSS emergence. In the discussion of the data, regarding the use of certain medications, more information of their mechanisms of action on cellular and molecular level can also be added.

Also, I have some technical remarks:

2.      The citations include some inaccessible (e.g. 1) and inappropriately (e.g. 6) sources.

Author Response

Manuscript ID: ijms-2567477
Type of manuscript: Review
Title: Beyond the umbrella: a systematic review of the interventions for prevention of and reduction in the incidence and severity of ovarian hyperstimulation syndrome in patients who undergo in vitro fertilization
treatments.
Authors: Stefano Palomba *, Flavia Costanzi, Scott M. M Nelson, Aris Besharat, Donatella Caserta, Peter Humaidan

Reviewer

Ovarian hyperstimulation syndrome (OHSS) is a serious medical condition, often due to the controlled ovarian stimulation during assisted reproductive technologies (ART) therapy. According to the available data, moderate to severe OHSS is observed in 1-5% of the cycles performed. Therefore, the manuscript treats a contemporary issue, and can be beneficial predominantly to ART practitioners. The authors present data for the effectiveness of the different approaches for prevention of and reduction in the incidence and severity of OHSS.

I accept the presented manuscript as a continuation of the beforehand published paper in 2023 of the same authors: (Palomba, S., Costanzi, F., Nelson, S.M. et al. Interventions to prevent or reduce the incidence and severity of ovarian hyperstimulation syndrome: a systematic umbrella review of the best clinical evidence. Reprod Biol Endocrinol 21, 67 (2023). https://doi.org/10.1186/s12958-023-01113-6).

Authors

We thank the Reviewer for the opportunity to revise the manuscript and have developed a point-by-point response below. We hope that the revised manuscript is now suitable for publication.

As the reviewer highlights the current manuscript builds on the previous umbrella review, which due to the nature of umbrella reviews was solely able to include interventions with previous systematic reviews.  The current paper includes all interventions which were identified through a robust search strategy but were not eligible for the umbrella review.

Reviewer

The methodology of the review and the statistical analysis are both well described. However, I have the following recommendations to the manuscript:

  1. As a whole the work is based on statistical analysis of existing data, there is too little to no molecular biology and I believe that the manuscript will be more appropriate to be published in clinical or endocrinological journal. Therefore, I think that in the introduction more information can be added regarding the molecular mechanisms for OHSS emergence. In the discussion of the data, regarding the use of certain medications, more information of their mechanisms of action on cellular and molecular level can also be added.

Authors

We thank the Reviewer for this excellent suggestion, and we have revised the introduction to the manuscript to include potential molecular mechanisms for OHSS. Similarly, we have now included detailed mechanisms of action for the drugs/strategies for OHSS prevention in the specific sections.

Reviewer

Also, I have some technical remarks:

  1. The citations include some inaccessible (e.g. 1) and inappropriately (e.g. 6) sources.

Authors

We have checked reference 1, and the link” https://www.rcog.org.uk/media/or1jqxbf/gtg_5_ohss.pdf” is correctly reported and the reference is available online. We have also added to the reference list “(31 August 2023 date last accessed)”.

We apologize for reference 6 which has now been deleted.

Reviewer 3 Report

/

Author Response

Manuscript ID: ijms-2567477
Type of manuscript: Review
Title: Beyond the umbrella: a systematic review of the interventions for prevention of and reduction in the incidence and severity of ovarian hyperstimulation syndrome in patients who undergo in vitro fertilization
treatments.
Authors: Stefano Palomba *, Flavia Costanzi, Scott M. M Nelson, Aris Besharat, Donatella Caserta, Peter Humaidan

Reviewer 3

---

Authors

As highlighted by the Editor, no comments were received by Reviewer 3.

Reviewer 4 Report

This is a mixture of different quality of studies, in an effort to find ways for preventing OHSS, with the exception of current meta-analyses on the subject.

The evidence is quite clear so far, through the existing papers. If authors wish to provide with new data/conclusions, a network meta-analysis or a IPD one, would be more meaningful.

Adequate. Minor edits are needed.

Author Response

Manuscript ID: ijms-2567477
Type of manuscript: Review
Title: Beyond the umbrella: a systematic review of the interventions for prevention of and reduction in the incidence and severity of ovarian hyperstimulation syndrome in patients who undergo in vitro fertilization
treatments.
Authors: Stefano Palomba *, Flavia Costanzi, Scott M. M Nelson, Aris Besharat, Donatella Caserta, Peter Humaidan

Reviewer

This is a mixture of different quality of studies, in an effort to find ways for preventing OHSS, with the exception of current meta-analyses on the subject.

The evidence is quite clear so far, through the existing papers. If authors wish to provide with new data/conclusions, a network meta-analysis or a IPD one, would be more meaningful.

Authors

We thank the Reviewers for the opportunity to revise the manuscript and have developed a point-by-point response below. We hope that the revised manuscript is now suitable for publication.

In particular, we thank the Reviewer for their suggestion regarding network meta-analysis or IPD meta-analysis. A “traditional” meta-analysis for prevention of OHSS has previously been undertaken (Palomba et al., 2023), and therefore the aim of the current manuscript was not to reassess this but rather highlight interventions that have not been well established by RCTs and subject to systematic review and secondly highlight the quality of their evidence. While we appreciate the potential value in an IPD meta-analyses, this would depend on the availability of individual patient data from the wide range of intervention undertaken and establishing appropriate data sharing agreements and protocols with all authors. To the regard of the network meta-analysis, those studies were excluded from final analysis at study design because greatly criticized from a methodological point of view (Li et al., 2011; Christofilos et al., 2022). In fact, indirect data may result in different conclusion when compared to data from direct comparisons (see references below). In addition, already recent network meta-analyses have been recently published. These points have already been discussed in our previous umbrella review of systematic reviews of RCTs with meta-analyses (Palomba et al., 2023). However, we have revised the discussion including a comment on network meta-analyses and on the results of the recent available papers (Marino et al., 2022; Wu et al., 2022). The references list has been updated accordingly.

Christofilos et al. World J Methodol 2022, 12, 92-98. doi: 10.5662/wjm.v12.i3.92.

Li et al. BMC Med 2011, 9, 79. doi: 10.1186/1741-7015-9-79.

Marino et al. J Assist Reprod Genet 2022, 39, 1583-1601. doi: 10.1007/s10815-022-02503-2.

Palomba et al. Reprod Biol Endocrinol 21, 67 (2023). https://doi.org/10.1186/s12958-023-01113-6

Rouse et al. Intern Emerg Med 2017, 12, 103-111. doi: 10.1007/s11739-016-1583-7.

Wu et al. Front Endocrinol (Lausanne) 2022, 13, 808517. doi: 10.3389/fendo.2022.808517.

Reviewer 5 Report

The manuscript “Beyond the umbrella: a systematic review of the interventions for prevention of and reduction in the incidence and severity of ovarian hyperstimulation syndrome in patients who undergo in vitro fertilization treatments” suggests a potential prevention method for a patient with ovarian hyperstimulation syndrome (OHSS). Unfortunately, the authors indicate that several treatments could reduce OHSS, despite not being supported by systematic reviews with homogeneity of the RCTs. However, at present, several problems in their method need to be carefully considered for a potential diagnostic method in the future. The authors should summarize alternatives to overcome the problems of their method.

I would like to ask two questions as major points.

Major point

1. In this review, the authors assessed and graded OHSS patients’s efficacy and evidence base without including them in systematic reviews with meta-analysis of randomized controlled trials (RCTs). With or without meta-analysis of RCTs, how much does it affect the evaluation? Also, what kind of approach is needed for clinical application in the future?

2. In the discussion section, it is better to add information about alternatives to overcome the problems of their method.

Minor points

1. Page 1, Line 20: Please amend “was identify” to “was identified”.

2. Page 2, Line 34: There are two in before “vitro fertilization”.

3. Page 2, Line 37: Please add “,” after “cases”.

Minor editing of English language required.

Please check the text again.

Author Response

Manuscript ID: ijms-2567477
Type of manuscript: Review
Title: Beyond the umbrella: a systematic review of the interventions for prevention of and reduction in the incidence and severity of ovarian hyperstimulation syndrome in patients who undergo in vitro fertilization
treatments.
Authors: Stefano Palomba *, Flavia Costanzi, Scott M. M Nelson, Aris Besharat, Donatella Caserta, Peter Humaidan

Reviewer

The manuscript “Beyond the umbrella: a systematic review of the interventions for prevention of and reduction in the incidence and severity of ovarian hyperstimulation syndrome in patients who undergo in vitro fertilization treatments” suggests a potential prevention method for a patient with ovarian hyperstimulation syndrome (OHSS). Unfortunately, the authors indicate that several treatments could reduce OHSS, despite not being supported by systematic reviews with homogeneity of the RCTs. However, at present, several problems in their method need to be carefully considered for a potential diagnostic method in the future. The authors should summarize alternatives to overcome the problems of their method.

Authors

We thank the Reviewer for the opportunity to revise the manuscript and have developed a point-by-point response below. We hope that the revised manuscript is now suitable for publication.

We are unsure what the Reviewer would like to summarize, in the current study we undertook a comprehensive systematic review (available on PROSPERO website), and only included interventions that were not eligible for that umbrella review. In fact, the study is a continuation of previous systematic umbrella review of available systematic reviews of RCTs with meta-analysis recently published (Palomba et al., Reprod Biol Endocrinol 2023; https://doi.org/10.1186/s12958-023-01113-6). Both studies are the results of a comprehensive research. In the first publication a total of 28 systematic reviews of RCTs with meta-analysis for 33 interventions were analyzed confirming that the GnRH-ant use with or without GnRH-a triggering, the progestin-primed ovarian stimulation (PPOS) protocol, the use of mild stimulation, the metformin co-treatment (during ovarian stimulation) and the dopamine agonists administration after ovulation triggering are all effective intervention for reducing the OHSS risk.

Reviewer

I would like to ask two questions as major points.

Major point

  1. In this review, the authors assessed and graded OHSS patients’s efficacy and evidence base without including them in systematic reviews with meta-analysis of randomized controlled trials (RCTs). With or without meta-analysis of RCTs, how much does it affect the evaluation? Also, what kind of approach is needed for clinical application in the future?

 Authors

The studies included in the current manuscript were not eligible to be included in the umbrella review as they were not subject to systematic review and meta-analyses, a prerequisite for inclusion in an umbrella review as per PRIOR guidelines. These interventions were not subject to such robust assessment clinically, means that the next step would be further randomized controlled trials or epidemiological methods to assess causality for example natural experiment designs to provide real world evidence of their utility.

Reviewer

  1. In the discussion section, it is better to add information about alternatives to overcome the problems of their method.

 Author

We have revised the discussion incorporating further information regarding alternatives.

Reviewer

Minor points

Page 1, Line 20: Please amend “was identify” to “was identified”.

Authors

Now corrected.

Reviewer

Page 2, Line 34: There are two in before “vitro fertilization”.

Authors

It is "in-vitro fertilization”, and so the two in would be correct but we missed the hyphen.

Reviewer

Page 2, Line 37: Please add “,” after “cases”.

Authors

Now corrected.

Round 2

Reviewer 1 Report

After reading the response given by the authors  and after carefully reading the corrected manuscript, I changed my previous objection and would reccommend to accept the manuscript for publication. In its current version, the rational for this umbrella narrative review is well substantiated. The methodology is clearly explained, the findins well presented and the conclusions well substantiated. It has potential to add to the current knowledge base when confronting the potential risk of OHSS.  

Author Response

Thanks so much for your positive comment and definitive decision.

Reviewer 5 Report

I think that the revised manuscript has been fundamentally improved and includes the contents requested by the referees and editorial team.

Minor editing of English language required.

Author Response

(The authors gave the same response as above.)
